# Convergence of Generalized Belief Propagation Algorithm on Graphs with Motifs

## Abstract

Belief propagation is a fundamental message-passing algorithm for numerous applications in machine learning. It is known that belief propagation algorithm is exact on tree graphs. However, belief propagation is run on loopy graphs in most applications. So, understanding the behavior of belief propagation on loopy graphs has been a major topic for researchers in different areas. In this paper, we study the convergence behavior of generalized belief propagation algorithm on graphs with motifs (triangles, loops, etc.) We show under a certain initialization, generalized belief propagation converges to the global optimum of the Bethe free energy for ferromagnetic Ising models on graphs with motifs.

## 1 Introduction

Undirected graphical models, also known as Markov Random Fields (MRF), provide a framework for modeling high dimensional distributions with dependent variables. Ising models are a special class of discrete pairwise graphical models originated from statistical physics. Ising models have numerous applications in computer vision Ravikumar et al. (2010), bio-informatics Marbach et al. (2012), and social networks Eagle et al. (2009). Explicitly, the joint distribution of Ising model is given by

$$\mathbb{P}(X) = \frac{1}{Z} \exp\left( \beta(\sum_i h_i X_i + \sum_{(i,j)} J_{ij} X_i X_j) \right), \tag{1}$$

where $\{X_i\}_i \in \{\pm 1\}^n$ are random variables valued in a binary alphabet (also known as "spins"), $J_{ij}$ represents the pairwise interactions between spin $i$ and spin $j$, $h_i$ represents the external field for spin $i$, $\beta = 1/T$ is the reciprocal temperature, and $Z$ is a normalization constant called *partition function*.

Historically, Ising models are proposed to study ferromagnetism. However, researchers find the computational complexity is the main challenge of performing sampling and inference on Ising models. In the literature, there are multiple ways to tackle the computational complexity. One of the ways are Markov-Chain Monte Carlo (MCMC) algorithms. A well-known example is Gibbs sampling, which is a special case of the Metropolis–Hastings algorithm. Basically Gibbs sampling samples a random variable conditioned on the distribution based on the previous samples. It can be shown that Gibbs sampling generates a reversible Markov chain of samples. Thus, the stationary distribution of the Markov chain is the desired joint distribution over the random variables, and it can be reached after the *burn-in period*. However, it is also well-known that Gibbs sampling will become trapped on multi-modal distribution. For example, Smith and Roberts (1993) and Mengersen (1996) show that when the joint distribution is bi-modal, the Gibbs sampling iterations may be trapped in one of the modes, reducing the probability of convergence.

Another popular way to go around the computational complexity is *variational methods*, which makes some approximation to the joint distribution. These methods usually turn the inference problem with respect to the approximate joint distribution into some non-convex optimization problem, and solve it either by the standard optimization methods, e.g, gradient descent, or by specialized algorithms like belief propagation. However, due to the non-convexity, those methods usually do not have theoretical guarantees that the solution converges to the global optimum.

Belief propagation (BP) is an effective numerical method for solving inference problems on graphical models. It was originally proposed by Pearl (2014) for tree-like graphs. Ever since it plays a fundamental role in numerous applications including coding theory Frey et al. (1998); Richardson

and Urbanke (2001), constraint satisfaction problems Achlioptas and Moore (2006), and community detection in the stochastic block model Decelle et al. (2011). It is well-known that belief propagation is only exact for a model on a graph with locally tree-like structures. The long haunting question is, theoretically how does belief propagation perform on loopy graphs.

We now describe the related work and our contributions.

**Related work and our contribution**

In a classic work, Yedidia et al. (2003) establishes the connection between belief propagation and the Bethe free energy. He shows that there is one-to-one correspondence between the fixed points of belief propagation and stationary points of the Bethe free energy. Following his work, it is known that the Bethe free energy at the critical points can be represented in terms of fixed point messages of belief propagation Montanari (2013). In a recent work, Koehler (2019) further studies the properties of Bethe free energy at the critical points, and shows for ferromagnetic Ising models, initialized with all one messages, belief propagation converges to the fixed point corresponds to the global maximum of the Bethe free energy. However, those theories consider either asymptotic locally tree-like graphs, or loopy graphs with simple edges. Real technological, social and biological networks have numerous short and large loops and other complex motifs, which lead to non-tree-like structures and essentially loopy graphs with hyper edges. Newman Newman (2009); Karrer and Newman (2010) and Miller (2009) independently propose a model of random graphs with arbitrary distributions of motifs. And Yoon et al. (2011) generalizes the Belief Propagation to graphs with motifs.

Our work builds on generalized belief propagation on graphs with motifs Yoon et al. (2011) and the convergence of belief propagation on ferromagnetic Ising models on loopy graphs with simple edges Koehler (2019). In this paper, we show for ferromagnetic Ising models on graphs with motifs, with all messages initialized to one, generalized belief propagation converges to the fixed point corresponds to the global maximum of the Bethe free energy.

## 2 ISING MODELS ON GRAPHS WITH MOTIFS

Let us introduce the concept of graphs with motifs. In graphs with motifs, each vertex belongs to a given set of motifs. As shown in Fig.1a , different motifs can be attached to vertex $i$: a simple edge $(i, j)$, a triangle, a square, a pentagon, and other non-clique motifs. Graphs with motifs can be viewed as hyper-graphs where motifs play a role of hyper-edges. And the number of specific motifs attached to a vertex is equal to hyper-degree with respect to the specific motifs. In this paper, for simplicity, we only consider simple motifs such as simple edges, and cliques.

Consider the Ising model with arbitrary order of interactions among vertices in each motif on a hyper-graph. Let $M_l(i)$ denote a cluster of size $l$ attached to vertex $i$, where vertices $j_1, j_2, \ldots, j_{l-1}$ together with $i$ form the motif. And let $X$ denote the random variable of spin configurations, the Hamiltonian of the model is

$$E(X) = -\sum_i h_i X_i - \sum_{(i,j)} J_{ij} X_i X_j - \sum_{(i,j,k)} J_{ijk} X_i X_j X_k - \sum_{(i,j,k,l)} J_{ijkl} X_i X_j X_k X_l - \cdots \quad (2)$$

where the first sum corresponds to the external fields at each vertex, the second sum corresponds to the pairwise interactions on simple edges, the third sum corresponds to the higher order interactions among spins in triangles, the fourth sum corresponds to the higher order interactions among spins in squares, and so on. As discussed in the previous section, most previous works focus on Ising models with pairwise interactions. In this paper, we are interested in Ising models with higher order interactions. For simplicity, we consider Ising models with only external fields and higher order interactions in triangles. Our derivation can be extended to more general cases.

Consider Ising models with only external fields and higher order interactions in triangles, the Hamiltonian of the model is

$$E(X) = -\sum_i h_i X_i - \sum_{(i,j,k)} J_{ijk} X_i X_j X_k, \quad (3)$$

where $(i, j, k)$ is a triangle, which can also be denoted as $M_3(i)$, $M_3(j)$, or $M_3(k)$.

By Boltzmann's law, the joint distribution is defined by

$$P(X) = \frac{1}{Z} e^{-\beta E(X)}, \tag{4}$$

where $Z$ is the *partition function*.

Throughout this paper, we focus on *ferromagnetic* Ising models, which is defined below

**Definition 1.** *An Ising model is ferromagnetic if $J_{ijk} \geq 0$ for all triangle motifs $(i, j, k)$ and $h_i \geq 0$ for all $i$.*

We introduce a intermediate message $\mu_{M_3(i)}$ from a motif $M_3(i)$ to spin $i$.

$$\mu_{M_3(i)}(X_i) = \frac{e^{\beta \lambda_{M_3(i)} X_i}}{2 \cosh \beta \lambda_{M_3(i)}}. \tag{5}$$

In the literature, different works have different definitions of messages. $\mu_{M_3(i)}$ is not the message definition we eventually work with in this paper, but it helps to understand the connections between different works. So, abusing the terminology a little bit, we call it 'intermediate message'.

By the definition of generalized Belief Propagation, the probability that spin $i$ is in a state $X_i$ is determined by the normalized product of incoming intermediate messages from motifs attached to spin $i$ and the external field factor $e^{\beta h_i X_i}$,

$$P_i(X_i) = \frac{1}{A} e^{\beta h_i X_i} \prod_{\{M_3(i)\}} \mu_{M_3(i)}(X_i), \tag{6}$$

where $A$ is a normalization constant. And the belief update rule is given by:

$$\mu_{M_3(i)}(X_i) = B \sum_{\{X_j = \pm 1\}} e^{-\beta E(M_3(i))} \prod_j \prod_{\{M_3(j) \neq M_3(i)\}} \mu_{M_3(j)}(X_j), \tag{7}$$

where $E(M_3(i))$ is an energy of the interaction among spins in the triangle $M_3(i)$, and $B$ is a normalization constant.

Multiplying Equation (7) by $X_i$ and summing over all spin configurations, we obtain an equation for the effective field $\lambda_{M_3(i)}$,

$$\tanh\left(\beta \lambda_{M_3(i)}\right) = \frac{1}{Z(M_3(i))} \sum_{\{X_i, X_{j_1}, \dots = \pm 1\}} X_i e^{-\beta \tilde{E}(M_3(i))}, \tag{8}$$

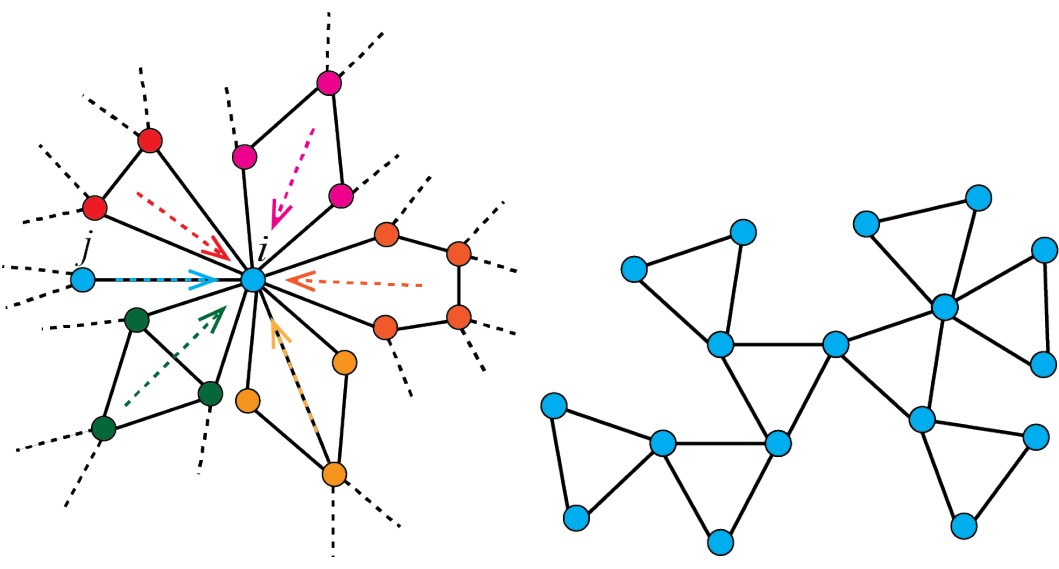

(a) Different motifs attached to vertex $i$       (b) Tree-like hyper-graph with triangle motifs only

Figure 1: Examples of hyper-graphs

where

$$\tilde{E}(M_3(i)) = -\sum_{n=1}^{2} \Lambda_t(j_n) X_{j_n} - J_{ij_1j_2} X_i X_{j_1} X_{j_2}, \tag{9}$$

$$\Lambda_t(j) = h_j + \sum_{\{M_3(j) \neq M_3(i)\}} \lambda_{M_3(j)}, \tag{10}$$

$$Z(M_3(i)) = \sum_{\{X_i, X_{j_1}, \ldots = \pm 1\}} e^{-\beta \tilde{E}(M_3(i))}. \tag{11}$$

For more detailed explanations of Equations (7) to (11), please refer to Yoon et al. (2011).

Now, define a message from a spin $i$ to motif $M_3(i)$ as $\nu_{i \to M_3(i)} = \tanh(\lambda_{M_3(i)})$. More specifically, if the motif $M_3(i)$ is a triangle $(i, j, k)$, the message can be alternatively represented as $\nu_{i \to M_3(i)} = \nu_{i \to jk} = \tanh(\lambda_{M_3(i)})$. From now on, let the reciprocal temperature $\beta = 1$, we can further simplify Equation (8) as

$$\nu_{i \to jk} = \tanh\left(h_i + \sum_{\{m,n\} \in \partial i \setminus \{j,k\}} \tanh^{-1}(\tanh(J_{imn}) \nu_{m \to in} \nu_{n \to im})\right), \tag{12}$$

where $\partial i$ denotes the motifs attached to spin $i$. Equation (12) is the consistency equation for fixed point hyper-edge messages $\nu_{i \to jk}^*$ of the generalized belief propagation. Alternatively, we denote Equation (12) as $\nu_{i \to jk} = \phi(\nu)_{i \to jk}$.

## 3 BETHE FREE ENERGY OF HIGHER ORDER ISING MODELS

In order to get the Bethe free energy of our higher order Ising model (3), we need to go through the Gibbs variational principle as Yedidia et al. (2003) did for standard Ising models with pairwise interactions. Let $P^*$ be a joint distribution defined by our model (4). If we have some approximate joint distribution $P$, from Gibbs variational principle, we can write Gibbs free energy as

$$G(P) = -\sum_x P(x)E(x) - \sum_x P(x)\log P(x) \tag{13}$$

$$= -U(P) + S(P), \tag{14}$$

where $U(P)$ is called the *average energy*, and $S(P)$ is the *entropy*.

We would like to derive a Gibbs free energy that is a function of both the one-node beliefs $P_i(x_i)$ and the three-node beliefs $P_{ijk}(x_i, x_j, x_k)$. The beliefs should satisfy the normalization conditions and the marginalization conditions. In other words, $P$ lies in the following polytope of locally consistent distributions

$$\sum_{x_j, x_k} P_{ijk}(x_i, x_j, x_k) = P_i(x_i) \quad \text{for all triangles } i, j, k$$

$$\sum_{x_i} P_i(x_i) = 1 \quad \text{for all } i \tag{15}$$

$$P_i(x_i), P_{ijk}(x_i, x_j, x_k) \geq 0 \quad \text{for all triangles } (i, j, k), \text{ and all } x_i, x_j, x_k$$

Because we only consider external fields and higher order interactions with triangles in our model, the one-node and three-node beliefs are actually sufficient to determine the average energy. For our model (3) and for any approximate joint probability $P$ such that one-node marginal probabilities are $P_i(x_i)$ and the three-node marginal probabilities are $P_{ijk}(x_i, x_j, x_k)$, the average energy will have the form

$$U(P) = -\sum_{(i,j,k)} \sum_{x_i, x_j, x_k} P_{ijk}(x_i, x_j, x_k) J_{ijk} x_i x_j x_k - \sum_i \sum_{x_i} P_i(x_i) h_i x_i \tag{16}$$

The average energy computed with the true marginal probabilities $P_i^*(x_i)$ and $P_{ijk}^*(x_i, x_j, x_k)$ will also have this form, so if one-node and three-node beliefs are exact, the average energy given by Equation (16) will be exact.

For computing the entropy, we usually need an approximation. We can compute the entropy exactly if we can explicitly express the joint distribution $P(x)$ in terms of the one-node and three-node beliefs. If our graph were tree-like hyper-graph with triangle motifs only (see Fig. 1b as an example), we can in fact do that. In that case, we can represent the joint probability distribution in the form

$$P(x) = \frac{\prod_{(i,j,k)} P_{ijk}(x_i, x_j, x_k)}{\prod_i P_i(x_i)^{q_i - 1}}, \tag{17}$$

where $q_i$ is the hyper-degree of node $i$.

Using Equation (17), we get the Bethe approximation to the entropy as

$$S_{\text{Bethe}}(P) = - \sum_{(i,j,k)} \sum_{x_i, x_j, x_k} P_{ijk}(x_i, x_j, x_k) \log P_{ijk}(x_i, x_j, x_k)$$

$$+ \sum_i (q_i - 1) \sum_{x_i} P_i(x_i) \log P_i(x_i) \tag{18}$$

Combining Equation (16) and (18), we obtain the Bethe free energy

$$G_{\text{Bethe}}(P) = -U(P) + S_{\text{Bethe}}(P) \tag{19}$$

$$= \sum_{(i,j,k)} J_{ijk} \mathbb{E}_{P_{ijk}}[X_i X_j X_k] + \sum_i h_i \mathbb{E}_{P_i}[X_i]$$

$$+ \sum_{(i,j,k)} H_{P_{ijk}}(X_i, X_j, X_k) - \sum_i (q_i - 1) H_{P_i}(X_i) \tag{20}$$

Notice when the hyper-graph is a tree, the Bethe free energy $G_{\text{Bethe}}(P)$ will have the correct functional dependence on the beliefs. And solving the optimization problem: maximizing $G_{\text{Bethe}}(P)$ over the polytope of locally consistent distribution (15) will give the true marginals. For loopy hyper-graphs, the Bethe free energy $G_{\text{Bethe}}(P)$ is only an approximation, which is the essence of the variational methods.

We can derive the BP equations from the first-order optimality conditions for the aforementioned optimization problem. In other words, we can verify that *a set of beliefs gives a BP fixed point in any hyper-graph if and only if they are stationary points of the Bethe free energy* for the generalized BP. To see this, we need to add Lagrange multipliers to $G_{\text{Bethe}}(P)$ to form a Lagrangian $L$. Let $\lambda_{i \to jk}(x_i)$ be a multiplier that enforces the marginalization constraint $\sum_{x_j, x_k} P_{ijk}(x_i, x_j, x_k) = P_i(x_i)$, and $\lambda_i$ be a multiplier that enforces the normalization of $P_i(x_i)$. So, the largrangian corresponding to the optimization problem is

$$\mathcal{L}(P, \lambda) = G_{\text{Bethe}}(P) + \sum_{(i,j,k), x_i} \lambda_{i \to jk}(x_i) \left( \sum_{x_j, x_k} P_{ijk}(x_i, x_j, x_k) - P_i(x_i) \right)$$

$$+ \sum_i \lambda_i \left( \sum_{x_i} P_i(x_i) - 1 \right) \tag{21}$$

where we ignore the constraints $P_i(x_i), P_{ijk}(x_i, x_j, x_k) \geq 0$ because, given other constraints, those constraints are always satisfied at a critical point.

The equation $\frac{\partial L}{\partial P_{ijk}(x_i, x_j, x_k)} = 0$ gives:

$$\log P_{ijk}(x_i, x_j, x_k) = J_{ijk} x_i x_j x_k + \lambda_{i \to jk}(x_i) + \lambda_{j \to ik}(x_j) + \lambda_{k \to ij}(x_k) - 1 \tag{22}$$

Setting $\lambda'_{i \to jk} = \frac{\lambda_{i \to jk}(1) - \lambda_{i \to jk}(-1)}{2}$, we find that at a critical point of the Lagrangian that

$$P_{ijk}(x_i, x_j, x_k) \propto \exp \left( J_{ijk} x_i x_j x_k + \lambda_{i \to jk}(x_i) + \lambda_{j \to ik}(x_j) + \lambda_{k \to ij}(x_k) \right) \tag{23}$$

$$\propto \exp \left( J_{ijk} x_i x_j x_k + \lambda'_{i \to jk} x_i + \lambda'_{j \to ik} x_j + \lambda'_{k \to ij} x_k \right) \tag{24}$$

And the equation $\frac{\partial L}{\partial P_i(x_i)} = 0$ gives:

$$(q_i - 1)(1 + \log P_i(x_i)) = \sum_{\{j,k\} \in \partial i} \lambda_{i \to jk}(x_i) - h_i x_i - \lambda_i \qquad (25)$$

Setting $\lambda'_{i \to jk} = \frac{\lambda_{i \to jk}(1) - \lambda_{i \to jk}(-1)}{2}$, we find that at a critical point of the Lagrangian that

$$P_i(x_i) \propto \exp\left( \frac{1}{q_i - 1} \sum_{\{j,k\} \in \partial i} \lambda_{i \to jk}(x_i) - \frac{h_i}{q_i - 1} x_i \right) \qquad (26)$$

$$\propto \exp\left( \frac{1}{q_i - 1} \sum_{\{j,k\} \in \partial i} \lambda'_{i \to jk} x_i - \frac{h_i}{q_i - 1} x_i \right) \qquad (27)$$

Furthermore, by differentiating with respect to $\lambda$, we see that the marginalization constraints are satisfied. Therefore, for any triangle $(i, j, k)$, $\sum_{x_j, x_k} P_{ijk}(x_i, x_j, x_k) = P_i(x_i)$. Hence,

$$P_i(x_i)^{q_i - 1} \propto \prod_{\{m,n\} \in \partial i \backslash \{j,k\}} \sum_{x_m, x_n} P_{imn}(x_i, x_m, x_n) \qquad (28)$$

$$\propto \sum_{x_{\partial i \backslash \{j,k\}}} \exp\left( \sum_{\{m,n\}} (J_{imn} x_i x_m x_n + \lambda'_{i \to mn} x_i + \lambda'_{m \to in} x_m + \lambda'_{n \to im} x_n) \right) \qquad (29)$$

$$= \exp(\sum_{\{m,n\}} \lambda'_{i \to mn} x_i) \sum_{x_{\partial i \backslash \{j,k\}}} \exp\left( \sum_{\{m,n\}} (J_{imn} x_i x_m x_n + \lambda'_{m \to in} x_m + \lambda'_{n \to im} x_n) \right) \qquad (30)$$

So

$$\exp(\lambda'_{i \to jk} x_i - h_i x_i) \propto \sum_{x_{\partial i \backslash \{j,k\}}} \exp\left( \sum_{\{m,n\}} (J_{imn} x_i x_m x_n + \lambda'_{m \to in} x_m + \lambda'_{n \to im} x_n) \right) \qquad (31)$$

$$\propto \prod_{\{m,n\} \in \partial i \backslash \{j,k\}} \sum_{x_m, x_n} \exp\left( J_{imn} x_i x_m x_n + \lambda'_{m \to in} x_m + \lambda'_{n \to im} x_n \right) \qquad (32)$$

Define $\nu_{i \to jk} := \tanh(\lambda'_{i \to jk})$, we have

$$\frac{1 + \nu_{i \to jk} x_i}{2} = \frac{e^{\lambda'_{i \to jk} x_i}}{e^{\lambda'_{i \to jk}} + e^{-\lambda'_{i \to jk}}} \qquad (33)$$

Let

$$f(x_i) = e^{h_i x_i} \prod_{\{m,n\}} \sum_{x_m, x_n} e^{J_{imn} x_i x_m x_n} e^{\lambda'_{m \to in} x_m + \lambda'_{n \to im} x_n} \qquad (34)$$

Then we see

$$\nu_{i \to jk} = \frac{f(1) - f(-1)}{f(1) + f(-1)} \qquad (35)$$

$$= \tanh\left( h_i + \sum_{\{m,n\} \in \partial i \backslash \{j,k\}} \tanh^{-1}(\tanh(J_{imn}) \nu_{m \to in} \nu_{n \to im}) \right) \qquad (36)$$

which is the BP consistency equation (12) we derived in Section 2.

Till this point, we represent the Bethe free energy in terms of beliefs corresponds to BP fixed points. In order to analyze the behavior of the Bethe free energy at BP fixed points, we need to represent the Bethe free energy in terms of the hyper-edge messages $\nu_{i \to jk}$, which is called *dual Bethe free energy* in the literature. First, we have the following lemma.

**Lemma 1.** *The dual Bethe free energy at a critical point can be defined by*

$$G^*_{Bethe}(\lambda) = \sum_i F_i(\lambda) - \sum_{(i,j,k)} F_{ijk}(\lambda), \tag{37}$$

*where*

$$F_i(\lambda) = \log \sum_{x_i} e^{h_i x_i} \prod_{\{m,n\} \in \partial i} \sum_{x_m, x_n} e^{J_{imn} x_i x_m x_n} e^{\lambda'_{m \to in} x_m + \lambda'_{n \to im} x_n} \tag{38}$$

$$F_{ijk}(\lambda) = \log \sum_{x_i, x_j, x_k} e^{J_{ijk} x_i x_j x_k + \lambda'_{i \to jk} x_i + \lambda'_{j \to ik} x_j + \lambda'_{k \to ij} x_k} \tag{39}$$

*Proof.* Recall the Bethe free energy

$$G_{\text{Bethe}}(P) = -U(P) + S_{\text{Bethe}}(P) \tag{40}$$

$$= \sum_{(i,j,k)} J_{ijk} \mathbb{E}_{P_{ijk}}[X_i X_j X_k] + \sum_i h_i \mathbb{E}_{P_i}[X_i] \tag{41}$$

$$+ \sum_{(i,j,k)} H_{P_{ijk}}(X_i, X_j, X_k) - \sum_i (q_i - 1) H_{P_i}(X_i) \tag{42}$$

By rearranging terms, we have

$$G_{\text{Bethe}}(P) = G^*_{\text{Bethe}}(\lambda) = \sum_i F_i(\lambda) - \sum_{(i,j,k)} F_{ijk}(\lambda), \tag{43}$$

where

$$F_i(\lambda) = \mathbb{E}[h_i X_i + \sum_{\{m,n\}} (J_{imn} X_i X_m X_n + \lambda'_{m \to in} X_m + \lambda'_{n \to im} X_n)] \tag{44}$$

$$+ \sum_{\{m,n\} \in \partial i} H(X_i, X_m, X_n) - (q_i - 1) H(X_i) \tag{45}$$

and

$$F_{ijk}(\lambda) = \mathbb{E}[J_{ijk} X_i X_j X_k + \lambda'_{i \to jk} X_i + \lambda'_{j \to ik} X_j + \lambda'_{k \to ij} X_k] + H(X_i, X_j, X_k) \tag{46}$$

W.l.o.g., let us look at the term $F_{ijk}(\lambda)$, let $f(X) = J_{ijk} X_i X_j X_k + \lambda'_{i \to jk} X_i + \lambda'_{j \to ik} X_j + \lambda'_{k \to ij} X_k$, it can be rewritten as

$$F_{ijk}(\lambda) = \mathbb{E}[f(X)] - \mathbb{E}[\log P_{ijk}(X_i, X_j, X_k)] \tag{47}$$

$$= \mathbb{E}[\log \frac{e^{f(X)}}{P_{ijk}(X_i, X_j, X_k)}] \tag{48}$$

From Equation (4), we know

$$P_{ijk}(X_i, X_j, X_k) = \frac{1}{Z_{ijk}} e^{f(X)}, \tag{49}$$

where $Z_{ijk}$ is a normalization constant $Z_{ijk} = \sum_{x_i, x_j, x_k} e^{f(X)}$. Substitute it back into Equation (47), we have

$$F_{ijk}(\lambda) = \mathbb{E}[\log(Z_{ijk})] = \log(Z_{ijk}) = \log(\sum_{x_i, x_j, x_k} e^{f(X)}) \tag{50}$$

$$= \log \sum_{x_i, x_j, x_k} e^{J_{ijk} x_i x_j x_k + \lambda'_{i \to jk} x_i + \lambda'_{j \to ik} x_j + \lambda'_{k \to ij} x_k} \tag{51}$$

$\square$

If we use the definition $\nu_{i \to jk} := \tanh(\lambda'_{i \to jk})$, and define $\theta_{ijk} = \tanh(J_{ijk})$, we have the following corollary:

**Corollary 1.** *The dual Bethe free energy in terms of hyper-edge messages is*

$$G^*_{Bethe}(\nu) = \sum_i F_i(\nu) - \sum_{(i,j,k)} F_{ijk}(\nu), \tag{52}$$

*where*

$$F_i(\nu) = \log\left[e^{h_i}\prod_{m,n}(1 + \theta_{imn}v_{m\rightarrow in}\nu_{n\rightarrow im}) - e^{-h_i}\prod_{m,n}(1 - \theta_{imn}v_{m\rightarrow in}\nu_{n\rightarrow im})\right] \tag{53}$$

$$F_{ijk}(\lambda) = \log\left(1 + \theta_{ijk}\nu_{i\rightarrow jk}\nu_{j\rightarrow ik}\nu_{k\rightarrow ij}\right) \tag{54}$$

## 4 OPTIMIZATION LANDSCAPE

Now, we can study the behavior of the Bethe free energy at critical points. The following lemma establishes that $\phi(\nu)_{i\rightarrow jk}$ is a concave monotone function for some non-negative $\nu$.

**Lemma 2.** *Suppose that* $f(x) = \tanh(h + \sum_{(i,j)}\tanh^{-1}(x_ix_j))$ *for any* $h \geq 0$. *Then* $f$ *is a concave monotone function on the domain* $[x^*, 1)^n$.

*Proof.* Observe that

$$\frac{\partial f}{\partial x_i}(x) = \frac{1 - f(x)^2}{1 - (x_jx_i)^2}x_j \geq 0, \tag{55}$$

which proves monotonicity, and

$$\frac{\partial^2 f}{\partial x_ix_k}(x) = \frac{1 - f(x)^2}{(1 - (x_jx_i)^2)(1 - (x_lx_k)^2)}\left[-2f(x)x_jx_l + \mathbb{1}(k = j, l = i)(1 + (x_ix_j)^2)\right.$$
$$\left. + \mathbb{1}(k = i, l = j)2(x_ix_j)^2\right]. \tag{56}$$

Note that for any non-negative vector $w$, if we let

$$w_i' = \frac{\sqrt{1 - f(x)^2}}{1 - (x_jx_i)^2}x_jw_i, \quad w_k' = \frac{\sqrt{1 - f(x)^2}}{1 - (x_kx_l)^2}x_lw_k \tag{57}$$

Then we have,

$$\sum_{i,k} w_i\frac{\partial^2 f}{\partial x_ix_k}w_k \tag{58}$$

$$= \sum_{i,k} w_i'\left[-2f(x) + \mathbb{1}(k = j, l = i)(\frac{1}{x_ix_j} + x_ix_j) + \mathbb{1}(k = i, l = j)2x_ix_j\right]w_k' \tag{59}$$

$$= -2f(x)(\sum_i w_i')^2 + \sum_{(ij)} w_i'w_j'(\frac{1}{x_ix_j} + x_ix_j) + \sum_i w_i'^2 2x_ix_j \tag{60}$$

$$\leq \sum_i -2(f(x) - x_ix_j)w_i'^2 + \sum_{(ij)} w_i'w_j'(\frac{1}{x_ix_j} + x_ix_j - 2f(x)) \tag{61}$$

$$= \sum_{(i,j)} w_i'w_j'\left[\frac{1}{x_ix_j} + x_ix_j - 2f(x)\left(1 + (1 - \frac{x_ix_j}{f(x)})(\frac{1}{q_i}\frac{w_i'}{w_j'} + \frac{1}{q_j}\frac{w_j'}{w_i'})\right)\right] \tag{62}$$

For any edge $(i, j)$, let $C = \frac{1}{q_i}\frac{w_i'}{w_j'} + \frac{1}{q_j}\frac{w_j'}{w_i'}$ (note $C \geq 2/\sqrt{q_iq_j}$), and

$$g(x) = \frac{1}{x} + x - 2f(x)\left(1 + C(1 - \frac{x}{f(x)})\right). \tag{63}$$

Due to the fact $x < f(x)$, we know $g(x) \to \infty$ as $x \to 0$, and $g(1) < 0$. Since $g(x)$ is continuous over $(0, 1)$, if we assume $x_{ij}^*$ is the largest root for $g(x)$ in $(0, 1)$, we know $g(x) < 0$ in $(x_{ij}^*, 1)$. Let $x^* = \max_{(i,j)} x_{ij}^*$, we have

$$\sum_{i,k} w_i \frac{\partial^2 f}{\partial x_i x_k} w_k \leq 0, \tag{64}$$

for $x \in [x^*, 1)^n$. □

We define the set of *pre-fixpoints* and *post-fixpoints* messages similar as in Koehler (2019):

$$S_{\text{pre}} = \{\nu : x^* \leq \phi(\nu)_{i \to jk} \leq \nu_{i \to jk}\}, \quad S_{\text{post}} = \{\nu : x^* \leq \nu_{i \to jk} \leq \phi(\nu)_{i \to jk}\} \tag{65}$$

From Lemma 2, we know $S_{\text{post}}$ is a convex set, while $S_{\text{pre}}$ is typically non-convex and even disconnected. Next, we show the gradient of the dual Bethe free energy is well-behaved on these sets:

**Lemma 3.** *If $\nu \in S_{pre}$ then $\nabla G_{Bethe}^*(\nu) \leq 0$ and if $\nu \in S_{post}$ then $\nabla G_{Bethe}^*(\nu) \geq 0$*

*Proof.* The lemma will follow if we compute the gradient of the dual Bethe free energy function $G_{\text{Bethe}}^*(\nu)$.

$$\frac{\partial G_{\text{Bethe}}^*(\nu)}{\partial \nu_{j \to ik}} = \frac{\partial F_i(\nu)}{\partial \nu_{j \to ik}} - \frac{\partial F_{ijk}(\nu)}{\partial \nu_{j \to ik}}$$

$$= \frac{e^{h_i} \theta \nu_{k \to ij} \prod_{m,n \in \partial i \setminus \{j,k\}}(1 + \theta \nu_{m \to in} \nu_{n \to im}) - e^{-h_i} \theta \nu_{k \to ij} \prod_{m,n \in \partial i \setminus \{j,k\}}(1 - \theta \nu_{m \to in} \nu_{n \to im})}{e^{h_i} \prod_{m,n}(1 + \theta v_{m \to in} \nu_{n \to im}) - e^{-h_i} \prod_{m,n}(1 - \theta v_{m \to in} \nu_{n \to im})}$$

$$- \frac{\theta \nu_{i \to jk} \nu_{k \to ij}}{1 + \theta \nu_{i \to jk} \nu_{j \to ik} \nu_{k \to ij}}$$

$$= \frac{1}{\nu_{j \to ik} + 1/(\theta \nu_{k \to ij} \phi(\nu)_{i \to jk})} - \frac{1}{\nu_{j \to ik} + 1/(\theta \nu_{k \to ij} \nu_{i \to jk})}. \tag{66}$$

Recall $\phi(\nu)_{i \to jk}$ is the updated message from spin $i$ to motif $\{j, k\}$ based on the current messages $\nu$. If $\nu \in S_{\text{pre}}$ or $S_{\text{post}}$, then the signs of the gradient of Bethe free energy are determined by Equation (66) as claimed. □

Based on Lemma 2 and 3, we can prove our main theorem.

**Theorem 1.** *Suppose that generalized BP is run from initial messages $\nu_{i \to jk}^{(0)} = 1$ and there is at least one fixed point in $[x^*, 1)^n$. The messages converge to a fixed point $\nu^*$ of the generalized BP equations such that for any other fixed point $\mu$, $\mu_{i \to jk} \leq \nu_{i \to jk}^*$. Furthermore*

$$G_{Bethe}^*(\nu^*) = \max_{\nu \in S_{post}} G_{Bethe}^*(\nu) \tag{67}$$

*Proof.* If there is at least one fixed point in $[x^*, 1)^n$, and the initialization is $\nu_{i \to jk}^{(0)} = 1$ for all hyper-edges $\{i, j, k\}$. For each iteration of Belief Propagation, $\nu^{(t)} = \phi(\nu^{(t-1)})$.

From Lemma 2, we know $\phi$ is monotonic on $[x^*, 1)^n$. So, $\nu^{(0)}, \nu^{(1)}, \ldots, \nu^{(t)}$ is a coordinate-wise decreasing sequence, which will converge to some fixed point. By monotonicity, we see that for any fixed point $\mu \in [x^*, 1)^n$, $\mu_{i \to jk} \leq \nu_{i \to jk}^{(t)}$ for all $t$. Hence, it holds for $\nu^*$ as well.

Finally, consider any other point $\nu \in S_{\text{post}}$, by convexity of $S_{\text{post}}$, we know that the line segment from $\nu$ to $\nu^*$ is entirely contained in $S_{\text{post}}$. By Lemma 3, we see that for any point $x$ on this interpolating line segment that $\nabla G_{\text{Bethe}}(\nu) \cdot (\nu^* - \nu) \geq 0$, and integrating over this line segment gives us $G_{\text{Bethe}}(\nu) \leq G_{\text{Bethe}}(\nu^*)$. □

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
