# OpenReview forum: "Convergence of Generalized Belief Propagation Algorithm on Graphs with Motifs"
_ICLR.cc/2022/Conference — ICLR 2022 Submitted_

### Official Review · Reviewer_F8Ar · 2021-10-18

**Correctness:** 1
**Technical Novelty And Significance:** 2
**Empirical Novelty And Significance:** 1
**Recommendation:** 1
**Confidence:** 5

**Main Review:**

Studying the convergence behavior of belief propagation and generalized belief propagation is an ongoing research topic, with many aspects still being not well understood. The paper tackles this important problem and provides a favorable initialization strategy for a special types of models. That being said, the novelty of the current paper is limited and the findings only apply to a restricted setting (more precisely, Theorem 1 is analogous to Lemma 3.3 in [Koehler]. The proofs are also analogous. While the authors claim that the results generalize to a more general case, this seems not to be the case); moreover, the notation and terminology is often imprecise (to the point of being misleading); finally, the paper fails to reference important related work -- both classical papers on belief propagation as well as recent advances. More details are presented below.

## Limitations
I completely understand that one has to make certain model assumptions to prove something formally. But this paper makes model assumptions for which it has already been established (at least for plain BP) that the most accurate fixed point is the one aligned with the local potentials ([Koehler] and [4]). Since the energy landscape is also well understood in this scenario (see for example [4] and [8]) it comes as no surprise that the convergence results of [Koehler] can be generalized. The result would be stronger if it would hold for any choice of regions (motifs). From the derivations presented, this does not seem to hold. Alternatively, making certain statements about the energy landscape of generalized BP for models with frustrations or arbitrary local potentials and how considering certain motifs might render the originally non-convex free energy convex would be of great interest.

## Notation/Terminology
The Bethe free energy is only defined over pairs of random variables. The generalization to larger groups of variables is named Kikuchi free energy.
It is repeatedly stated that Ising models with an external field are considered but the equations always consider local fields. The major difference is that local fields are potentially different for every variable, i.e., we have $h_i$, whereas an external field is equivalent over all variables, i.e., we have $h_i=h$
The summation indices $(i,j), (i,j,k),...$  are used imprecise. One must ensure that every pair, triplet,... contributes only once to the energy (or normalize the energy accordingly).
The ferromagnetic model is not used consistently throughout the literature. Whereas [1] refers to ferromagnetic models whenever $J_{ij}>0$, [6] has the same definition as in this paper.
Equation 5 uses $\lambda(\ldot)$ that has never been introduced.
I cannot make sense of the double product in Equation 7. I would expect that the first product should go over all neighbors of $i$ (except for $M3(j)$) but not over all variables as stated here.

## Related Work
The generalization of the Bethe free energy to higher models in Section 3 is not novel. This is precisely the Kikuchi free energy.
The derivations in Section 3 are a straightforward generalization of the classical derivations for the Bethe free energy in [2].
The claim that there are not guarantees on how close the beliefs at the "best" fixed point is to the exact one are not entirely correct (see [5] and [7]).
The accumulation of the effect from multiple variables into one message seems similar to the concept of the effective field in [5].
Minor Remarks
The terms "variable" and "spin" are used interchangeably. Sticking to one form would improve the clarity.
In line 72 $X$ denotes the random variable. In the remaining paper it is always $X_i$
.
line 86: Ising models, which is defined -> which are defined
Please fix the references. [Koehler] for example has been published at this very conference (NeurIPS) and is not just an arXiv preprint.
[1] Mooij, Joris, and Hilbert Kappen. "Sufficient conditions for convergence of loopy belief propagation." (2012).

[2] Heskes, Tom. "On the uniqueness of loopy belief propagation fixed points." (2004).

[3] Meltzer, Talya, Amir Globerson, and Yair Weiss. "Convergent message passing algorithms-a unifying view." (2009).

[4] Knoll, Christian, Adrian Weller, and Franz Pernkopf. "Self-Guided Belief Propagation--A Homotopy Continuation Method." (2018).

[5] Knoll, Christian, and Franz Pernkopf. "Belief Propagation: Accurate Marginals or Accurate Partition Function–Where is the Difference?." (2020).

[6] Saade, Alaa, Florent Krzakala, and Lenka Zdeborová. "Spectral clustering of graphs with the bethe hessian." (2014).

[7] Ihler, Alexander, John Fisher, and Alan Willsky. "Loopy Belief Propagation: Convergence and Effects of Message Errors"

[8] Zachary Pitkow, Yashar Ahmadian, Ken Miller. "Learning unbelievable probabilities"

**Summary Of The Paper:**

The paper considers attractive models with unidirectional local potentials and studies a special case of generalized belief propagation, namely over triplets of graphs. For this model class, the paper generalizes the convergence results from [Koehler].

**Summary Of The Review:**

The paper generalizes the findings of [koehler] to one specific and restricted setting. The technical results are derived analogously to [koehler] and it is not clear how (or if) the findings would generalize; neither to more general potentials nor to more general versions of generalized belief propagation.
Moreover, the paper has sever issues regarding clarity and deviates from the classical literature in its clarity. Also, the related work is not cited adequately.

---

### Official Review · Reviewer_PMGJ · 2021-10-25

**Correctness:** 1
**Technical Novelty And Significance:** 1
**Empirical Novelty And Significance:** 1
**Recommendation:** 1
**Confidence:** 4

**Main Review:**


I have a suspicion that the proof of Lemma 2 of this paper is completely wrong.
I would like the author to clarify the following points.

1. The function $f$ is defined on a domain in n- dimensional space and the value is scalar.
   Is the sum i is over 1 to n and the sum j is also over 1 to n.
   Is this correct? (this just a sanity check.)

2. First of all what is your policy on proving concavity?
   Eq (64) does not imply concavity because w is limited to non-negative elements.
   (recall that non-negative matrices are not necessarily positive semi-definite)

3. The summation over $j$ is missing in Eq 55, and the summation over $j$ and $l$ are missing in Eq 56.
    (or do you abbreviate them? if so I recommend not do it because it is confusing.)

4. Again, Eq 57 look strange because the right hand side includes index j (resp. l)
   but the left hand shows no dependency on j (resp l).
   If you are omitting the sigma over j, I suggest you not do that because it is just confusing.

5. In Eq 60, the third term has explicit dependency over index j.
   This look wrong because the first and the second terms and Eq 58 has no dependency on j.

6. Can you explain how do you obtain Eq 61 from 60?

7. Between Eq 62 and 63, the definition of C depends on indices i and j.
   I recommended to use the notation $C_{ij}$ instead of $C$.
   Likewise, I recommended to use the notation $g_{ij}$ instead of $g$.

8. Is x in Eq 63 a scalar or a vector?
   If it is a vector, how should I understand $1/x$?
   If it is a scalar, I recommend to use other characters, say $t$, to avoid confusion.

9. I believe the author tries to show each summand of Eq 62 is negative.
   However I cannot obtain the summand by putting $x$ or $x_i x_j$ into g.

10. Between Eq 63 and 64, what do you mean by $x < f(x)$? Is $x$ a vector or scalar?


**Summary Of The Paper:**

The author considers a special class of attractive models and extends the convergence results of [Koehler]. In general, the convergence behavior of belief propagation algorithm is hard to be understood, however, attractive models are special class of models known to have simple optimization loss landscape. Therefore, the result of the paper is unlikely to be generalized to non-attractive models, limiting the application of the paper.

**Summary Of The Review:**

The main result of the paper, Theorem 1, is dependent on Lemma 2. Therefore, I would like to vote for rejection unless every single step of the proof of Lemma 2 is made clear.

---

### Official Review · Reviewer_BLzF · 2021-11-02

**Correctness:** 3
**Technical Novelty And Significance:** 2
**Empirical Novelty And Significance:** Not applicable
**Recommendation:** 3
**Confidence:** 3

**Main Review:**

This paper studies the Ising model, which is perhaps one of the most popular undirected graphical models (also known as Markov Random Fields). The authors study the belief propagation for solving inference problems on graphical models. It is known that BP produces exact solution on trees and locally tree-like graphs. The main contribution of this paper is to study the convergence behavior of BP on loopy graphs.

The authors study the Ising model on graphs with motifs. For such models, every vertex is associated with motifs that can be edges, triangles, squares, etc. The Hamiltonian is given by summing over all motifs and for each motif taking a product of spins. One can also view such a model as an Ising model on a hypergraph. The authors give detailed proofs to show that the generalized BP run from the initial all one message converges to a fixed point whose Bethe free energy is maximum. In fact, in the proof of this paper the authors only consider triangle motifs, so the Hamiltonian is a sum over all triangles. It is reasonable to infer that the same proof approach works for larger motifs as well.

One weakness of this paper is that the latter half of the paper is mainly proofs based and it is a little bit hard to follow the intuitions of results. Also, the main convergence result does not seem to say anything about the time it takes for the generalized BP to convergence. This could be disappointing to some people for application reasons as I suspect that the convergence time would increase with the size of the motifs, and that the convergence result might not be so meaningful for motifs with non-constant sizes. In that sense, and also given the known convergence results of BP for Ising (Koehler’19), the main result of this paper is not so impressive.


**Summary Of The Paper:**

This paper studies the convergence behavior of the generalized belief propagation algorithm on graphs with motifs. The authors show that when initialized as the all one message, the generalized belief propagation will converge to some fixed point that maximizes the Bethe free energy.


**Summary Of The Review:**

This is an interesting paper but I hope the authors can improve their presentations of their main results by removing some proofs to the supplementary materials and include more intuitions and novelty compared to previous works like Koehler’19. Also it is reasonable to include a study on the convergence rate of generalized BP, either theoretical or experimental.

---

### Official Review · Reviewer_bjzY · 2021-11-07

**Correctness:** 3
**Technical Novelty And Significance:** 2
**Empirical Novelty And Significance:** Not applicable
**Recommendation:** 3
**Confidence:** 4

**Main Review:**

Clarity:  The paper contains a number of typos, but they do not affect understanding.  I found the presentation of the results to be counterintuitive and frustrating, with a number of definitions presented out of order.  Further the notation is incredibly cumbersome in places. e.g., M_3(i), even though there are perfectly good and much more well-established notational conventions that could (and should) be adopted.  Finally, a number of results are presented as if they are novel for this work, e.g., eqns 21-36, which is confusing as they are established results that could simply be included with appropriate citations.

Novelty:  The final theoretical result appears to be novel, but the authors do a poor job of explaining which pieces are novel here or how this work relates to the previously established proof techniques, e.g., is this just a rehashing of the same arguments in a slightly more general case?

Significance:  While the result may be of interest, the proof techniques may possess only limited novelty and the practical applications are unclear.

Specific comments:

- The citation style should be fixed.  Use (author year) instead of author (year) for parenthetical citations.
- The notation in equation 3 is really challenging.  Is the sum over all possible 3 vertex combinations?  How do you distinguish two different triangles containing vertex i with the notation M_3(i)?
- Why is h_i assumed to be greater than or equal to zero?  This isn't the same as the binary ferromagnetic case?
- lambda is used in (5) before it is defined.  In fact, (5) is a complete non-sequitur and the presentation of the message passing scheme in general is difficult to follow.
- I don't understand the notation phi(v_{i\rightarrow jk).  It seems to be the same as (12) and only creates extra confusion in the discussion as it it several pages before it is used again.
- Are (21)-(36) and Lemma 1 standard results?  I think this needs a citation at minimum.
- How does Lemma 2 differ from the approach in previous work?  Indeed, how does anything here differ from the previous approach?

**Summary Of The Paper:**

The authors extend a result for BP on ferromagnetic Ising models to BP on higher order Ising models.  This is a purely theoretical result.

**Summary Of The Review:**

An interesting result that seems to rehashing a significant amount of existing work without citation, is hampered by a poor presentation, and is not careful to place itself in the context of existing results.

---

### Decision · Program_Chairs · 2022-01-20

**Decision:**

Reject

**Comment:**

All reviewers are very critical about the submitted paper regarding novelty of results, insufficient placement with respect to existing results, and clarity of presentation. The authors also did not submit a rebuttal. Hence I am recommending rejection of the paper.